# Is Less Sedentary Behavior, More Physical Activity, or Higher Fitness Associated with Sleep Quality? A Cross-Sectional Study in Singapore

**DOI:** 10.3390/ijerph17041337

**Published:** 2020-02-19

**Authors:** Robert A. Sloan, Youngdeok Kim, Susumu S. Sawada, Akihiro Asakawa, Steven N. Blair, Eric A. Finkelstein

**Affiliations:** 1Department of Social and Behavioral Medicine, Kagoshima University Graduate Medical School, Kagoshima 890-8520, Japan; asakawa@m2.kufm.kagoshima-u.ac.jp; 2Department of Kinesiology and Health Sciences, Virginia Commonwealth University, Richmond, VA 23284, USA; kimy13@vcu.edu; 3Faculty of Sport Sciences, Waseda University, Saitama 359-1192, Japan; s-sawada@waseda.jp; 4Exercise Science Arnold School of Public Health, University of South Carolina, Columbia, SC 29208, USA; sblair@mailbox.sc.edu; 5Lien Centre for Palliative Care, Duke-NUS Medical School, Singapore 169857, Singapore; eric.finkelstein@duke-nus.edu.sg

**Keywords:** cardiorespiratory fitness, sedentary, physical activity, sleep quality, combined association

## Abstract

Objectives: To examine the independent, joint, and fully combined associations of sedentary behavior (SB), moderate-to-vigorous physical activity (MVPA), and cardiorespiratory fitness (CRF) with the odds of poor sleep quality (SQ). Methods: We performed a secondary data analysis on 757 working adults (male = 345) in Singapore, with an average age of 35.2 years. The Pittsburgh Sleep Quality Index was used to assess SQ. Objectively measured MVPA and SB were each obtained using an accelerometer. A non-exercise prediction equation was used to estimate CRF. Logistic regression models were used to determine associations. Results: In total, 13.2% of the sample (n = 100) was identified as having poor SQ. After adjusting for study covariates, independent analyses revealed a clear inverse association for higher CRF and lower odds of poor SQ (OR = 0.50; 95% CI = 0.28–0.91). SB and MVPA demonstrated no independent associations. Joint associations revealed that odds of having poor SQ for those with low CRF was higher regardless of SB level and was further deteriorated by lower MVPA in the fully combined model. The fully combined model also demonstrated that those with lower SB, higher MVPA, and higher CRF had the lowest odds of having poor SQ (OR = 0.28; 95% CI = 0.10–0.78). Conclusions: Physical activity/exercise training programs that aim to improve CRF may be useful in lowering the odds or poor SQ in working adults.

## 1. Introduction

The World Health Organization recently identified poor sleep quality (SQ) as a public health problem that increases the risk of premature morbidity and mortality [1]. Poor SQ is associated with poorer cardiovascular health, metabolic health, brain health and accidents [2]. A broad definition of SQ is the subjective perception of one’s whole sleep experience [3]. A recent investigation in Singapore demonstrated poor SQ to be associated with lower health-related quality of life in working adults [4]. The adverse health impact and economic costs of poor SQ are a significant public health concern in industrialized societies such as Singapore, whereby the country was found to have the shortest sleep duration out of 20 countries [5,6]. Though clinical interventions are useful for improving SQ in patients with diagnosed sleep disorders, they do not provide a public health solution for the more pervasive health problem of poor SQ. Given the ubiquity of the health problem, health promotion strategies aimed at reducing poor SQ may be a worthwhile investment. 

Meta-analyses and systematic reviews suggest that moderate-to-vigorous physical activity (MVPA) moderately improves SQ in adults [3]. MVPA can be characterized by an energy expenditure ≥ 3 absolute METs while participating in leisure time, occupation/housework, or commuting activities [3]. There is also evidence to suggest that sedentary behavior (SB) and cardiorespiratory fitness (CRF) are each associated with sleep health [7,8]. However, it is unclear whether, and to what extent being less sedentary, more physically active or more fit may be associated with SQ.

Each health behavior characteristic (SB, MVPA, and CRF) has been independently associated with obesity, diabetes, high blood pressure, heart disease, cancer, and all-cause mortality [9]. However, public health guidelines tend to primarily focus on physical activity. Therefore, researchers have recently called for all three characteristics to be included when developing guidelines [9]. To date, no studies have examined all three characteristics (SB, MVPA, and CRF) in this context of SQ. 

Sedentary behavior has been defined as any waking behavior characterized by a low energy expenditure [≤1.5 times resting metabolic rate (i.e., METs)] while sitting, reclining, or lying down [3]. Due to differences in study quality and methodology, and the limited amount of studies, findings for the relationship with SB and SQ are inconclusive. A meta-analysis found SB to be associated with an increased risk of insomnia and sleep disturbance but not daytime sleepiness and poor SQ in adults [10]. More recent results from observational studies suggest an independent relationship with SB and SQ [7,11,12]. 

CRF is a comprehensive marker of exercise/physical activity patterns, heredity, age, smoking, diet, body composition, and health status [3,13]. There is emerging evidence from large observational studies that low CRF may be a risk factor for sleep-related disorders independent of MVPA or body mass index [8,14,15,16]. Uchida et al. have hypothesized that the chronic effect of exercise on sleep may be mediated through improvements in fitness. The authors further stated, “These benefits likely reflect the summed and interacting benefits of numerous physiological alterations by regular exercise, which directly and indirectly affect SQ” [17]. Lastly, no studies have investigated the joint relationships of CRF and SB with SQ. Comparing the various groupings of exposures (i.e., fit/sedentary/low active or unfit/non-sedentary/active) may better help elucidate the risk associated with poor SQ. A more comprehensive understanding may help develop new hypotheses regarding mechanisms and help establish interventions targeted at improving SQ in working adults.

To our knowledge, no studies have investigated the comprehensive interplay between SB, MVPA, and CRF with SQ. Therefore, the purpose of the current study was to: (1) examine the independent associations between objectively measured SB, MVPA, and estimated CRF with subjective SQ in generally healthy working adults in Singapore; (2) examine the joint and fully combined associations of all three exposures with SQ.

## 2. Method 

### 2.1. Data and Sample

Using baseline data from our original randomized controlled trial, this secondary data analysis examined the cross-sectional associations of SB, MVPA, and CRF with SQ [18]. Adults aged between 21 and 64 years were recruited by convenience sampling across thirteen worksites in Singapore. During the original investigation, 1307 participants were assessed for inclusion—of which, 507 were excluded. The details of the original exclusion method can be found elsewhere [18]. After exclusion, the original sample consisted of 800 full-time employees who were English speaking, not pregnant, and determined to be able to participate in MVPA. All eligible employees completed an informed consent form and underwent objectively measured BMI, blood pressure, and resting heart rate. Participants were also requested to complete an online baseline questionnaire on socio-demographic information, SQ, psychological distress, and other self-reported chronic health conditions (Diabetes, high blood cholesterol, high blood pressure, heart disease, chronic obstructive pulmonary disease, cancer, depression, osteoporosis, arthritis, asthma, stroke, kidney ailments). Based on the Pittsburgh Sleep Quality Index, we verified that the full-time employees did not work night shifts. 

Baseline MVPA and SB data were assessed for one week via accelerometry. The institutional review board at the National University of Singapore approved the study protocol, and the experimental protocol of the original study was registered at ClinicalTrials.gov, number NCT01855776. Detailed information about the original study, including the descriptions of the baseline dataset has been published extensively elsewhere [18]. For this study, 21–64-year-olds who provided valid measures of study outcome variables were included (43 participants were excluded). The final analytic sample consisted of 757 adults (male = 345), with an average age of 35.5 years (SD = 8.5). The post-hoc power analysis was performed to justify the sample size for this secondary data analysis. Based on the observed odds ratio (OR) of 2.2 of reporting poor SQ among individuals with lower CRF when compared to those with higher CRF (groups will be defined later), it was estimated that the sample size of 757 was sufficient to achieve approximately 90% statistical power at an alpha level of 0.05.

### 2.2. Measures

#### 2.2.1. Pittsburgh Sleep Quality Index (PSQI)

The PSQI has been validated in both clinical and non-clinical populations and is widely used as a health outcome variable in epidemiological and clinical investigations. The index contains seven components reflected over a 4-week interval: subjective SQ, sleep latency, sleep duration, habitual sleep efficiency, sleep disturbances, the use of sleep medication, and daytime dysfunction. A global score is determined from 0 to 21 by assessing 19 items on a 4-point Likert scale. Global scores >5 are indicative of overall poor SQ and are used to categorize the participants into poor and good SQ groups [19].

#### 2.2.2. Cardiorespiratory Fitness (CRF)

While maximal oxygen uptake (max VO_2_ ) is typically measured in a laboratory setting to determine the CRF level, non-exercise algorithms have demonstrated to be useful in epidemiological investigations [13]. We used a valid non-exercise algorithm to estimate max VO_2_ and classify CRF [20]. The algorithm uses self-reported exercise level, sex, age, objectively measured resting heart rate and BMI, to determine max VO_2_. The non-exercise algorithm used in our study has been demonstrated to have a high correlation with laboratory-measured max VO_2_ (r = 0.83) [20]. Non-exercise algorithms have also been found to correctly classify the CRF level with a high rate of accuracy (>90%) and predict health outcomes [13]. Max VO_2_ was classified into CRF tertiles (<33.3, <37.7, and ≥37.7 mL/kg/min). After that, we dichotomized CRF into ‘Higher’ (upper tertile) and ‘Lower’ (middle and lower tertiles) categories [21]. BMI was calculated using the objectively measured height (Seca 217 Mobile Stadiometer, Seca Deutschland, Hamburg, Germany) and weight (Seca 869 Mobile floor Scale, Seca GmBH, Hamburg, Germany). Resting heart rate was measured using the Welch Allyn Spot Vital Signs Blood Pressure monitor in the seated position.

#### 2.2.3. Accelerometer Data

Participants were asked to wear the ActiGraph GT3X+ accelerometer (ActiGraph, Pensacola, FL, USA) attached to an elastic belt on their waist during waking hours for seven consecutive days. The accelerometer data were downloaded in 60 sec epoch using the ActiLife software (version 8.0.0) and processed using the “accelerometry” package (version 2.2.4) in an open-source R program (version 3.1.2; R Core Team, Vienna, Austria). Non-wear time was examined using the Choi’s algorithm [22], and the time spent in SB (<1.5 METs) and MVPA (≥3.0 METs) during wear time were extracted based on a vertical axis threshold of <100 and a vector magnitude threshold of ≥2690 counts per minute, respectively [23,24]. Daily SB and MVPA minutes were averaged across valid days, defined as days with ten or more hours of wear time. Average SB and MVPA minutes per day were further adjusted for average wear time using a least square method, and participants were dichotomized into the two categories where individuals in low and mid tertiles (<8.50 h/day for SB; and <52.12 min/day for MVPA) were assigned into ‘Lower’ and individuals in upper tertile (≥8.50 h/day for SB; ≥52.12 min/day for MVPA) were assigned into ‘Higher’ groups for the respective variable.

#### 2.2.4. Other Variables

Study covariates included self-reported age, sex, ethnicity, marital status, chronic conditions, psychological distress, and weekly consumption of sugar-sweetened beverages. Chronic condition status was classified into none or one or more conditions. BMI was objectively measured, and categorization was based on the World Health Organization report for Asians (<23, 23–26.9, ≥27 kg/m^2^) [25]. Non-specific psychological distress was determined by using the Kessler Psychological Distress Scale (K10). The K10 uses ten 5-point Likert scale questions about anxiety and depressive symptoms experienced in the most recent 4 weeks to determine a summary score. Global scores, ≥ 20, indicated psychological distress [26]. Further details of these measures have been reported in earlier research and can found in previous publications [18]. 

#### 2.2.5. Statistical Analysis

Descriptive statistics were calculated for study variables. Differences in categorical and continuous variables between SQ groups were examined by *x*^2^ test of independence and independent Sample *t*-test, respectively. The bivariate correlations between each pair of ST, MVPA, and CRF were examined. Three sets of binary logistic models were tested to examine the independent association of SB, MVPA, and CRF with the likelihood of having poor SQ. Model 1 included SB, MVPA, or CRF as a single predictor in separate models to estimate unadjusted association; model 2 included all three predictors in a single model to estimate the mutually adjusted association between study variables, and model 3 additionally adjusted for study covariates to estimate the fully adjusted association with SQ. Three additional logistic regression models were established to examine the joint associations of SB, MVPA, and CRF with the likelihood of having poor SQ. Each model included a joint category created for each pair of study variables (i.e., model 1: SB/MVPA; model 2: SB/CRF; and model 3: MVPA/CRF) in addition to the variable that is not combined and study covariates. Lastly, eight separate group categorizations were established to examine the fully combined associations of SB/MVPA/CRF with the likelihood of having poor SQ. All associations were presented by the OR and 95% confidence interval (CI). The correlations between each pair of SB, MVPA, and CRF were estimated for testing multicollinearity among independent variables (Appendix A). There was no multicollinearity (r < 0.70) present in the models [27]. We also tested for interaction effects between CRF and BMI on SQ and found no significant interaction effects. All data analyses were conducted using the SAS v9.4 (SAS Institute, Cary, NC, USA).

## 3. Results 

The demographic characteristics of the study sample are presented in Table 1. Descriptive statistics of key outcome variables along with components of PSQI are available in Appendix A. The prevalence of poor SQ in our sample was 13.2% (N = 100). The *x*^2^ test of independence showed that SQ was significantly associated with ethnicity, chronic medical conditions, psychological distress, and consumption of sweetened, energy/sports, and regular soft drinks (*P*’s < 0.05). A relatively large portion of individuals with poor SQ were of Chinese ethnicity (58.8%), had one or more medical conditions (26.8%) and higher psychological distress (33.0%), and more frequently consumed sweetened/energy (18.6%) or regular soft drinks (15.5%) when compared to those for individuals with good SQ. 

Table 2 presents the unadjusted, mutually, and fully adjusted ORs and 95% CIs estimating independent associations of SB, MVPA, and CRF with SQ. The results showed that SB and MVPA levels are not significantly associated with SQ across all models, where the odds of having poor SQ was not significantly different between individuals in the lower and higher ST or MVPA groups. Higher CRF was consistently associated with having lower odds of poor SQ across the models. Individuals in the higher CRF group had lower odds of having poor SQ (unadjusted OR = 0.46) when compared to those with lower CRF group. After adjusting study covariates, individuals in the higher CRF group had significantly lower odds of having poor SQ (fully adjusted OR = 0.50; 95% CI = 0.28–0.91).

The results of multiple logistic regression analyses examining the joint associations of each pair of MVPA, SB, and CRF with SQ are presented in Table 3. The model, including joint SB/MVPA, showed that the odds of having poor SQ were not significantly different between groups. When joint SB and CRF were considered, the lower SB/higher CRF category was the referent; SB groups with lower CRF demonstrated greater odds of having poor SQ regardless of SB level (OR = 2.71 for higher SB and lower CRF; and OR = 2.89 for lower SB and lower CRF). The model examining the joint association of MVPA and CRF demonstrated that compared to the referent higher MVPA/higher CRF group, the lower MVPA/lower CRF group showed greater odds of having poor SQ (OR = 3.02; 95% CI = 1.25–7.29). The fully combined association model of SB, MVPA, and CRF with SQ is presented in Table 4. When comparing to the lower SB/lower MVPA/lower CRF (i.e., L/L/L) as a referent group, the lower SB/higher MVPA/higher CRF group (i.e., L/H/H) as well as the lower SB/higher MVPA/lower CRF group (i.e., L/H/L) showed lower odds of having poor SQ (OR = 0.28; 95% CI = 0.10–0.78 for L/H/H; and OR = 0.29; 95% CI = 0.09–0.88). Additionally, when comparing to the L/H/L and L/H/H groups, the higher SB/lower MVPA/lower CRF group (i.e., H/L/L) had greater odds of having poor SQ (OR = 3.29; 95% CI = 1.04–10.35 when comparing to the L/H/L; and OR = 3.36; 95% CI = 1.18–9.54 when comparing to L/H/H groups). 

## 4. Discussion 

This study aimed to examine the independent, joint, and fully combined associations of objectively measured SB, MVPA, and estimated CRF with self-reported SQ in a generally healthy working adult population aged 21–64 years old. Our main findings are: (1) only higher CRF was independently associated lower odds of poor SQ; (2) in addition to having higher CRF, those with concomitant higher MVPA and lower SB levels had lower risk; and (3) *conversely,* those with lower CRF, lower MVPA, and high SB levels had a higher risk. To the best of our knowledge, this is the first study to report the comprehensive interplay of SB, MVPA, and CRF with the likelihood of poor SQ.

Contrary to our findings, studies have collectively demonstrated that MVPA is associated with SQ [3]. A potential reason is the domain of MVPA primarily used in meta-analyses and reviews was *exercise training* (defined as planned, a structured, repetitive, or intentional movement intended to improve CRF) [28,29]. Moreover, the accepted absolute accelerometry MVPA threshold used in our secondary dataset may have overestimated MVPA in younger adults [30]. It is also important to point out that accelerometry only accounts for ambulatory activity while CRF provides a better indication of all types of MVPA (e.g., swimming, biking, rowing).

In line with our estimated CRF findings, observational studies using laboratory measured CRF have shown independent inverse associations with higher CRF and lower insomnia, sleep apnea, and difficulty sleeping [8,14,15,16]. Although there is some evidence to suggest reducing weight may improve SQ, results from meta-analyses suggest that exercise improves SQ independent of weight change. While our data indicated that those who were overweight/obese had about an 11% higher prevalence of poor SQ, we found BMI to be a minimal confounder and it did not moderate the observed associations of CRF with SQ [29]. Correspondingly, recent meta-analyses of RCTs in sleep apnea patients showed that exercise training concomitantly improves CRF and reduces sleep apnea severity and improves SQ without significant changes in BMI in adults who get low amounts of MVPA [31,32]. 

Our findings regarding joint MVPA/CRF were somewhat in line with Zou et al., who found that higher insomnia risk was consistently associated with lower CRF across lower levels of MVPA [14]. We found lower CRF combined with lower MVPA levels to be associated with poor SQ. Unlike Zou et al., we did not see increased risk in the lower CRF/higher MVPA category. This difference may be due to the differences in the populations, sleep outcomes, and methodology.

There is limited literature regarding the relationship between SB and SQ. In line with our findings, a recent meta-analysis indicated no independent associations between SB and SQ but did demonstrate associations with other sleep outcomes [10]. More recent results from two large observational studies in generally healthy adults indicated that when the self-reported SB was high, the odds of reporting poor SQ/sleep problems nearly doubled [11,12]. Loprinzi et al. examined data (N = 5563) from a large US sample of adults using objectively measured SB [7]. Independent of MVPA, the researchers found that for every 1 h increase in sedentary time, individuals were 16% and 22% more likely to feel unrested and sleepy during the day. The researchers also suggested the possibility of a bidirectional relationship was at play, being more sleepy increased sedentary time. 

There is some evidence for joint associations of SB and MVPA with SQ. Farnsworth et al. found that adults with high levels of MVPA in conjunction with moderate levels of sedentary behavior had lower odds of sleep problems and sleep disorders [33]. Conversely, we found adults not to have an increased risk of poor SQ when joint associations of MVPA/SB levels were examined. The differences are likely because of the methodology and sleep outcome used in our investigation. However, the best evidence to challenge our SB/MVPA findings of the association was revealed in a recent RCT. Edwards and colleagues found that when exercise training was removed from habitually active adults, and SB was increased after one week, there was a significant decrease in SQ [34]. 

Markedly, no studies to date have investigated the association of the simultaneous or fully combined associations of SB/MVPA/CRF with SQ or other sleep outcomes. Our findings that only CRF was associated with SQ seem to support the previously mentioned hypothesis by Uchida et al. [17]. In addition, there appeared to be a compounding effect on lowering risk or increasing risk when MVPA and SB levels were considered in combination with CRF. These unique findings may add to future physical activity guidelines [3]. 

### 4.1. Mechanisms

Though SB, MVPA, and CRF have been linked to SQ, the physiological mechanisms are not well understood. Researchers have suggested that exercise affects the central and somatic nervous systems to improve SQ. Fatigue and altered thermoregulation may account for the improvement in sleep duration and slow-wave sleep [14,17]. SQ improvement through fitness is thought to occur because of improved parasympathetic control [17]. Also, higher CRF results in better oxygen utilization and transport, which may intern promote better SQ [14]. It has been suggested that better SQ may occur because of the positive psychosocial adjustments regular exercise brings about through increases in the brain-derived neurotrophic factor [35]. Higher CRF has also been associated with better stress management, which may impact SQ [36]. Lastly, there is some evidence to suggest that SB may impact SQ through the nocturnal rostral fluid shift [37].

### 4.2. Limitations

This investigation is not without limitations. First, because of the cross-sectional design, interpretations about causality between SB, MVPA, CRF, and SQ cannot be made. Therefore, a bidirectional relationship between the independent and dependent variables cannot be ruled out. Second, the SQ outcome was based on self-reported recall over the last 30 days. Third, smoking status was not assessed. Fourth, accelerometry cannot account for non-ambulatory forms of exercise such as biking, swimming, skating, or high-intensity interval training. This gap may have lead to an underestimation of MVPA in our dataset. The absolute accelerometry MVPA threshold used in our secondary dataset may have overestimated MVPA in some individuals leading to possible misclassification [30]. Longer follow-up beyond 7 days with ActiGraph may have reduced the variance. Fifth, max VO_2_ estimation equations may underestimate and overestimate CRF at the upper and lower ends of the distribution. However, they have a high rate of accuracy for correctly classifying individuals into high- and low-fit categories. Nonetheless some individuals may have been misclassified [13]. Lastly, the age range and ethnicity/race of the study population may limit the generalizability of the results.

## 5. Conclusions

Our findings show that only higher fitness was independently associated with SQ. In addition to having higher fitness, reducing sedentary behavior and increasing physical activity may provide further benefits. Health promotion programs may consider focusing on MVPA/exercise interventions that aim to improve CRF. Further experimental studies should examine the impact of measured CRF, SB, and MVPA with sleep outcomes along with basic research focused on causal mechanisms.

## Figures and Tables

**Table 1 ijerph-17-01337-t001:** Demographic Characteristics of the Participants.

	Total	Pittsburgh Sleep Quality Index (PSQI) ^a^	
		Good Sleep Quality	Poor Sleep Quality
**N (%)**	757	657 (86.8%)	100 (13.2%)
Age (N, %)			
20–39.9 years	537 (70.9%)	458 (69.7%)	79 (79.0%)
40–64.9 years	220 (29.1%)	199 (30.3%)	21 (21.0%)
Gender (N, %)			
Male	345 (45.6%)	306 (46.6%)	39 (39.0%)
Female	412 (54.4%)	351 (53.4%)	61 (61.0%)
Ethnicity (N, %)			
Chinese	531 (70.1%)	472 (71.8%)	59 (59.0%)
Others	226 (29.9%)	185 (28.2%)	41 (41.0%)
Marital status (N, %)			
Married	420 (55.5%)	372 (56.6%)	52 (52.0%)
Others	337 (44.5%)	285 (43.4%)	48 (48.0%)
Chronic medical conditions ^b^ (N, %)			
One or more	121 (16.0%)	94 (14.3%)	27 (27.0%)
None	636 (84.0%)	563 (85.7%)	73 (73.0%)
Psychological distress (n, %) ^c^			
Yes	72 (9.5%)	40 (6.1%)	32 (32.0%)
No	685 (90.5%)	617 (93.9%)	68 (68.0%)
Body mass index ^d^ (N, %)			
<23 kg/m^2^	335 (44.3%)	300 (45.7%)	35 (35.0%)
23–26.99 kg/m^2^	256 (33.8%)	220 (33.5%)	36 (36.0%)
≥27 kg/m^2^	166 (21.9%)	137 (20.8%)	29 (29.0%)
Consumption of sugary/energy drinks (N, %)			
≤twice per week	429 (56.7%)	384 (58.5%)	45 (45.0%)
3–4 times per week	241 (31.8%)	204 (31.0%)	37 (37.0%)
≥5 times per week	87 (11.5%)	69 (10.5%)	18 (18.0%)
Consumption of regular soda (N, %)			
≤twice per week	446 (58.9%)	401 (61.0%)	45 (45.0%)
3–4 times per week	231 (30.5%)	191 (29.1%)	40 (40.0%)
≥5 times per week	80 (10.6%)	65 (9.9%)	15 (15.0%)

Values are presented as n (%) for all variables. ^a^ The categorization of participants into good and poor sleep quality was based on total scores of the PSQI (the descriptive statistics of PSQI scores are presented in Appendix A). ^b^ Chronic medical conditions included diabetes, high blood cholesterol, hypertension, heart disease, chronic obstructive pulmonary disease, and stroke. ^c^ Psychological distress was measured by the Kessler Psychological distress Scale and categorization was based on the cut-off scores.

**Table 2 ijerph-17-01337-t002:** Independent Associations of SB, MVPA and CRF with Sleep Quality.

	Model 1 ^a^	Model 2 ^b^	Model 3 ^c^
MVPA (min/day) ^d^			
Lower	1.00 (referent)	1.00 (referent)	1.00 (referent)
Higher	0.75 (0.47–1.20)	0.88 (0.54–1.42)	0.62 (0.35–1.10)
SB (hrs/day) ^d^			
Lower	(referent)	(referent)	(referent)
Higher	1.14 (0.74–1.77)	1.09 (0.70–1.71)	1.15 (0.70–1.89)
CRF (MaxVO_2_) ^d^			
Lower	(referent)	(referent)	(referent)
Higher	0.46 * (0.27–0.76)	0.47 * (0.28–0.79)	0.50 * (0.28–0.91)

MVPA = moderate-to-vigorous intensity physical activity. SB = sedentary behavior. CRF = cardiorespiratory fitness. Values are odds ratios and 95% confidence intervals estimated from logistic regression models predicting the likelihood of having poor sleep quality. ^a^ Model 1 included single independent variable. ^b^ Model 2 included all independent variables. ^c^ Model 3 included all independent variables in addition to study covariates including age, gender, ethnicity, marital status, chronic medical conditions, body mass index, psychological distress, and consumptions of sweetened drinks and soda. ^d^ Higher and lower represent the 3rd and 1st + 2nd tertile groups, respectively, for each study variable. * *P* ≤ 0.05.

**Table 3 ijerph-17-01337-t003:** Two-Way Combined Associations of SB, MVPA and CRF with Sleep Quality.

SB (hrs/day) and MVPA (min/day)			SB (hrs/day) and CRF (MaxVO_2_)			MVPA (min/day) and CRF (MaxVO_2_)		
Joint Group ^a^	N (%)	OR (95% CI)	Joint Group ^b^	N (%)	OR (95% CI)	Joint Group ^c^	N (%)	OR (95% CI)
H and L	198 (26.2%)	1.88 (0.94–3.76)	H and L	175 (23.1%)	2.71 * (1.16–6.30)	H and L	133 (17.6%)	1.74 (0.68–4.46)
H and H	55 (7.3%)	1.7 (0.45–3.53)	H and H	78 (10.3%)	2.23 (0.83–5.93)	H and H	119 (15.7%)	1.00 (referent)
L and L	307 (40.6%)	1.68 (0.86–3.25)	L and L	329 (43.5%)	2.89 * (1.31–6.39)	L and L	371 (49.0%)	3.02 * (1.25–7.29)
L and H	197 (26.1%)	1.00 (referent)	L and H	175 (23.1%)	1.00 (referent)	L and H	134 (17.7%)	1.39 (0.51–3.75)

MVPA = moderate-to-vigorous intensity physical activity. SB = sedentary behavior. CRF = cardiorespiratory fitness. Values are odds ratio and 95% confidence interval estimated from logistic regression models predicting the likelihood of having poor sleep quality after controlling for study covariates (age, gender, ethnicity, marital status, chronic medical conditions, body mass index, psychological distress, and consumptions of sweeten and soft drinks) in addition to SB, MVPA, or CRF depending on the models. ^a^ Groups in the left and right side indicate SB and MVPA groups, respectively, where Higher (H) and lower (L) represent the 3rd and 1st + 2nd tertile groups, respectively. ^b^ Groups in the left and right side indicate SB and CRF groups, respectively, where Higher (H) and lower (L) represent the 3rd and 1st + 2nd tertile groups, respectively. ^c^ Groups in the left and right side indicate MVPA and CRF groups, respectively, where Higher (H) and lower (L) represent the 3rd and 1st + 2nd tertile groups, respectively. * *P* ≤ 0.05.

**Table 4 ijerph-17-01337-t004:** Fully-Combined Associations of SB, MVPA and CRF with Sleep Quality.

Combined Group (SB-MVPA-CRF) ^a^	N (%)	Odds Ratios (95% Confidence Intervals) ^b^						
H-H-H	25 (3.3%)	0.57 (0.14–2.22)	0.97 (0.23–4.02)	1.97 (0.38–10.12)	2.01 (0.43–9.35)	0.60 (0.15–2.40)	1.00 (0.18–5.55)	0.73 (0.16–3.33)
H-H-L	53 (7.0%)	0.77 (0.30–1.98)	1.33 (0.47–3.78)	2.69 (0.72–10.12)	2.75 (0.82–9.30)	0.82 (0.31–2.17)	1.37 (0.33–5.74)	1.00 (reference)
H-L-H	30 (4.0%)	0.57 (0.16–2.00)	0.97 (0.26–3.62)	1.97 (0.41–9.53)	2.01 (0.47–8.69)	0.60 (0.17–2.17)	1.00 (referent)	
H–L-L	145 (19.2%)	0.94 (0.51–1.75)	1.62 (0.72–3.62)	3.29 * (1.04–10.35)	3.36 * (1.18–9.54)	1.00 (referent)		
L-H-H	94 (12.4%)	0.28 * (0.10–0.78)	0.48 (0.16–1.44)	0.98 (0.25–3.82)	1.00 (referent)			
L-H-L	81 (10.7%)	0.29 * (0.09–0.88)	0.49 (0.15–1.66)	1.00 (referent)				
L-L-H	103 (13.6%)	0.58 (0.28–1.24)	1.00 (referent)					
L-L-L	226 (39.9%)	1.00 (referent)						

MVPA = moderate-to-vigorous intensity physical activity. SB = sedentary behavior. CRF = cardiorespiratory fitness. The estimates were from logistic regression model predicting the likelihood of reporting poor sleep quality after controlling for study covariates (age, gender, ethnicity, marital status, chronic medical conditions, body mass index, psychological distress, and consumptions of sweeten and soft drinks). ^a^ H and L represent the higher (3rd tertile) and lower (1st + 2nd tertile groups) groups for SB, MVPA, and CRF groups. ^b^ The odds ratio estimates reported in each column indicates the differences in odds of reporting poor sleep quality between the group in the respective row and the reference group in that column. * *P* ≤ 0.05.

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
