# Peer review of "Is Less Sedentary Behavior, More Physical Activity, or Higher Fitness Associated with Sleep Quality? A Cross-Sectional Study in Singapore"

_ijerph, 2020, doi:10.3390/ijerph17041337_

Round 1

Reviewer 1 Report

This was a clear and well-written secondary data analysis.

Indicating the secondary analysis nature of this research would be important to include in the abstract and early in the methods section.  

Although BMI is a problematic measure of obesity, I would like to know how obesity interacts with CRF and sleep quality.

Author Response

RR Attached

Reviewer 2 Report

Comments

The work was performed at a high methodological level, the authors collected a sufficient amount of experimental material and used adequate methods of statistical data processing. I have a few comments, mainly regarding methods and discussion of results.

The authors write: “The detail of the original exclusion method can be found elsewhere [16].” But I did not find in this article information about whether people working on night shifts participated in the study. If such people participated in the study, then the authors must either include this variable in the model, or indicate this in the limitations section. The work (Martinez-Gomez et al. 2010. Int. J. Obesity, 34 (10), 1501-1507.) shows that there is a significant correlation between MVPA and CRF, so the risk of collinearity in models that include these indicators is quite high. Therefore, the authors should provide more detailed information about how they evaluated multicollinearity and what values ​​of indicators that characterize it they received. In the discussion section, it is necessary to indicate one of the possible mechanisms of the protective action of CRF on sleep quality - by reducing stress-related symptoms of depression (Gerber, et al., 2013. Patient education and counseling, 93 (1), 146-152.)

Author Response

RR attached. 

Reviewer 3 Report

The authors of the work presented an important topic.

I suggest to deepen the Introduction / background of the problem.

Author Response

RR attached, 

Reviewer 4 Report

INTRODUCTION: in my opinion this section could be expanded to better describe helth related issue about sedentary behaviour, bad sleep quality and cardiorespiratory fitness.

METHOD: 

PSQI is not a gold standard for sleep evaluation. Polysomnography is the gold standard. Have you taken into consideration that PSQI evaluates the last 30-day, while with actigraph you evaluated the last 7-day? maybe data from these two type of surveys  are not superimposable. You used a nonexercise algorith to evaluate CFR, but among your large sample there could be sportsmen or people who practice physical activity every day. Maybe you can not generalize or you need to use a more specific algorithm. Why have you not ask about physiscal activity habits? you used a standars treshold for the actigraph counts evaluation. However the age of your sample is very wide and such a treshold could not fitfrom 20 to 65 yaers old. Are you able to repeat the analysis with another and more age specific treshold?

DISCUSSION: do you think that with your analysis and discussion did you answer to question expressed in the title? From your discussion in really difficult to sum up the positive or negative implications of your study.

Author Response

RR attached. 

Round 2

Reviewer 4 Report

The changes applied are sufficient and exhaustive.

No more comments from my side.